# Pruning Edges and Gradients to Learn Hypergraphs from Larger Sets

**David W. Zhang**
University of Amsterdam
w.d.zhang@uva.nl

**Gertjan J. Burghouts**
TNO
gertjan.burghouts@tno.nl

**Cees G. M. Snoek**
University of Amsterdam
cgmsnoek@uva.nl

## Abstract

This paper aims for set-to-hypergraph prediction, where the goal is to infer the set of relations for a given set of entities. This is a common abstraction for applications in particle physics, biological systems, and combinatorial optimization. We address two common scaling problems encountered in set-to-hypergraph tasks that limit the size of the input set: the exponentially growing number of hyperedges and the run-time complexity, both leading to higher memory requirements. We make three contributions. First, we propose to predict and supervise the *positive* edges only, which changes the asymptotic memory scaling from exponential to linear. Second, we introduce a training method that encourages iterative refinement of the predicted hypergraph, which allows us to skip iterations in the backward pass for improved efficiency and constant memory usage. Third, we combine both contributions in a single set-to-hypergraph model that enables us to address problems with larger input set sizes. We provide ablations for our main technical contributions and show that our model outperforms prior state-of-the-art, especially for larger sets.

## 1 Introduction

In many applications we are given a set of entities and want to infer their relations, including vertex reconstruction in particle physics [1, 2], inferring higher-order interactions in biological and social systems [3, 4] or combinatorial optimization problems, such as finding the convex hull or Delaunay triangulation [2, 5]. The broad spectrum of different application areas underlines the generality of the abstract task of *set-to-hypergraph* [2]. Here, the hypergraph generalizes the pairwise relations in a graph to multi-way relations, a.k.a. hyperedges. In biological systems, multi-way relationships (hyperedges) among genes and proteins are relevant for protein complexes and metabolic reactions [6]. A subgroup in a social network can be understood as a hyperedge that connects subgroup members [7] and in images interacting objects can be modeled by scene graphs [8] which is useful for counting objects [9]. We distinguish the set-to-hypergraph task from the related but different task of link prediction that aims to discover the missing edges in a graph, given the set of vertices *and a subset of the edges* [10]. For the set-to-hypergraph problem considered in this paper, we are given a new set of nodes without any edges at inference time.

A common approach to set-to-hypergraph problems is to decide for *every* edge, whether it exists or not [2]. For a set of $n$ nodes, the number of all possible hyperedges grows in $\mathcal{O}(2^n)$. Thus, simply storing a binary indicator for every hyperedge already becomes intractable for moderately sized $n$. This is the *scaling* problem of set-to-hypergraph prediction that we will address in this paper. Common combinatorial optimization problems introduce the additional challenge of *super-linear time complexity*. For example, convex hull finding in $d$ dimensions has a run time complexity of $\mathcal{O}(n \log(n) + n^{\lfloor \frac{d}{2} \rfloor})$ [11]. This means that larger input sets require more computation regardless of the quality of the hypergraph prediction algorithm. Indeed, it was observed in [2] that for larger set sizes performance was worse. In this paper, we aim to address the scaling and complexity problems in order to predict hypergraphs for larger input sets.

D. W. Zhang et al., Pruning Edges and Gradients to Learn Hypergraphs from Larger Sets. *Proceedings of the First Learning on Graphs Conference (LoG 2022)*, PMLR 198, Virtual Event, December 9–12, 2022.

**Figure 1: Iterative hypergraph refinement framework.** To reduce the memory complexity, we represent the hypergraph by its incidence matrix and base the loss function solely on existing edges (Section 2). Problems of arbitrary complexity can be addressed without an additional memory footprint via iterative refinement and skips in the backward pass (Section 3). We propose a simple recurrent neural network that refines the edge and node features, based on wich it predicts the incidence matrix while preserving the permutation symmetries (Section 4).

**Our main contributions are:**

1. We propose a set-to-hypergraph framework that represents the hypergraph structure with a *pruned incidence matrix* — the incidence matrix without the edges (rows) that have no incident nodes (Section 2). Our framework improves the asymptotic memory requirement from $\mathcal{O}(2^n)$ to $\mathcal{O}(mn)$, which is linear in the input set size. We prove that pruning the edges does not affect the learning dynamics.

2. To address the complexity problem, we introduce in Section 3 a training method that encourages iterative refinement of the predicted hypergraph with memory requirements scaling constant in the number of refinement steps. This addresses the need for more compute by complex problems in a scalable manner.

3. We combine in Section 4 the efficient representation from the first contribution with the requirements of the scalable training method from the second contribution in a recurrent model that performs iterative refinement on a pruned incidence matrix. Our model handles different input set sizes and varying numbers of edges while respecting the permutation symmetry of both. Figure 1 illustrates the three contributions, the neural network, and the organization of the paper.

4. In our experiments in Section 5, we provide an in-depth ablation of each of our technical contributions and compare our model against prior work on common set-to-hypergraph benchmarks.

## 1.1 Preliminary

Hypergraphs generalize standard graphs by replacing edges that connect exactly two nodes with hyperedges that connect an arbitrary number of nodes. Since we only consider the general version, we shorten hyperedges to edges in the remainder of the paper. For the set-to-hypergraph task, we are given a dataset $\mathcal{D} = \{(\boldsymbol{X}, \boldsymbol{C})_i\}_{i \in [1\ldots N]}$ of tuples consisting of input node features $\boldsymbol{X} \in \mathbb{R}^{n \times d}$ and the target $\boldsymbol{C}$ that specifies the connections between the nodes. The target can be represented with a collection of $k$-dimensional adjacency tensors for $k \in [2 \ldots n]$. The $k$-dimensional adjacency tensor is the generalization of the 2-dimensional adjacency matrix to $k$-edges which are incident to $k$ different nodes. Alternatively, we can represent the connections between the nodes with the incidence matrix $\boldsymbol{I}^* \in \{0, 1\}^{m \times n}$, which we adopt and motivate in this work. Thus, our goal is to learn a model $f : \mathbb{R}^{n \times d} \to \{0, 1\}^{m \times n}$ that maps a set $\boldsymbol{X}$ of $d_{\boldsymbol{X}}$-dimensional node features to the incidence matrix of the corresponding hypergraph. Note that we treat two edges as the same if they are incident to the same nodes. In particular, edges do not differ by any attributes or weights.

An important baseline is the Set2Graph neural network architecture [2]. We focus on it because most previous architectures follow a similar structure. In Set2Graph, $f$ consists of a collection of functions $f \coloneqq (F^1, F^2, \ldots, F^k)$, where $F^k$ maps a set of nodes to a set of $k$-edges. All $F^k$ are composed of three steps: 1. a set-to-set model maps the input set to a latent set, 2. a broadcasting step forms all possible $k$-tuples from the latent set elements, and 3. a final graph-to-graph model that predicts for each $k$-edge whether it exists or not. The output of every $F^k$ is then represented by an adjacency tensor, a tensor with $k$ dimensions each of length $n$. Serviansky et al. [2] show that this can approximate any continuous set-to-$k$-edge function, and by extension, the family of $F^k$ functions can approximate any continuous set-to-hypergraph function. Since the asymptotic memory scaling of

$F^k$ is in $\mathcal{O}(n^k)$, modeling $k$-edges beyond $k > 2$ becomes intractable in many settings and one has to apply heuristics to recover higher-order edges from pairwise edges [2].

## 2 Scaling by pruning the non-existing edges

In this section, we address the memory scaling problem encountered in set-to-hypergraph tasks. We propose a framework for training a model to only predict the positive edges, that is edges that have at least one incident node. At training time, we prune all the *non-existing edges* (zero-valued rows in the incidence matrix) which significantly reduces the memory complexity when the hypergraph is sufficiently sparse. We refer to the resulting incidence matrix as a *pruned incidence matrix*. Since it does not contain all possible edges, a row is no longer associated with a specific edge. Ideally, the model does not rely on a specific order of the edges or the nodes. We ensure this by predicting an entry in the incidence matrix $\boldsymbol{I}^*_{i,j}$ solely based on the latent representations of the node $i$ and edge $j$. For now, we assume that we are given these representations and discuss the specific neural network architecture that produces them in the following sections. In what follows, we describe the representation of the predicted hypergraph and the loss function.

**Nodes.** Each row in the input $\boldsymbol{X}$ corresponds to the input features of a node in the hypergraph. Additionally, we denote by $\boldsymbol{V} \in \mathbb{R}^{n \times d_{\boldsymbol{V}}}$ the latent features of the nodes.

**Edges.** The set of all possible edges can be expressed using the power set $\mathcal{P}(\boldsymbol{V}) \setminus \{\varnothing\}$, that is the set of all subsets of $\boldsymbol{V}$ minus the empty set. Different from the situation with the nodes, we do not know which edge will be part of the hypergraph, since this is what we want to predict. Listing out all $2^{|\boldsymbol{V}|}-1$ possible edges and deciding for each edge whether it exists or not, becomes intractable very quickly. Furthermore, we observe that in many cases the number of existing edges is much smaller than the total number of possible edges. We leverage this observation by keeping only a fixed number of edges $m$ that is sufficient to cover all (or most) hypergraphs for a given task. Thus, we improve the memory requirement from $\mathcal{O}(2^{|\boldsymbol{V}|}d_{\boldsymbol{E}})$ to $\mathcal{O}(md_{\boldsymbol{E}})$, where $d_{\boldsymbol{E}}$ is the dimension of the latent edge representation $\boldsymbol{E} \in \mathbb{R}^{m \times d_{\boldsymbol{E}}}$. All possible edges that do not have an edge representation in $\boldsymbol{E}$ are *implicitly* predicted to not exist.

**Incidence matrix.** Since both the nodes and edges are represented by latent vectors, we require an additional component that explicitly specifies the connections. Different from previous approaches, we use the incidence matrix over adjacency tensors [2, 12]. The two differ in that incidence describes whether an edge is connected to a node, while adjacency describes whether an edge between a subset of nodes exists. The rows of the incidence matrix $\boldsymbol{I}^*$ correspond to the edges and the columns to the nodes. We treat the prediction task as a binary classification per entry in the incidence matrix. Thus the predicted incidence probability matrix $\boldsymbol{I} \in [0,1]^{m \times n}$ specifies the probabilities of the $i$-th edge being incident to the $j$-th node. In principle, we can express any hypergraph in both representations, but pruning edges becomes especially simple in the incidence matrix, where we just remove the corresponding rows. We interpret the pruned edges that have no corresponding row in the pruned $\boldsymbol{I}$ as having zero probability of being incident to any node.

**Loss function.** We apply the loss function only on the incidence probability matrix $\boldsymbol{I}$. For efficiency purposes, we would like to train each incidence value separately as a binary classification and apply a constant threshold $(> 0.5)$ on $\boldsymbol{I}$ at inference time to get a discrete-valued incidence matrix. In probabilistic terms, this translates to a factorization of the joint distribution $p(\boldsymbol{I}^*|\boldsymbol{X})$ as, $\prod_{i,j} p(\boldsymbol{I}^*_{i,j}|\boldsymbol{X})$. In order to retain the expressiveness of modeling interrelations between different incidence probabilities, we impose a requirement on the model $f$: the probability $\boldsymbol{I}_{i,j}$ produced by $f$ depends on the latent feature vectors $\boldsymbol{E}_i$ and $\boldsymbol{V}_j$ that we assume to already capture information of their incident nodes and edges respectively. We could alternatively factorize $p(\boldsymbol{I}^*|\boldsymbol{X})$ through the chain rule, but that is much less efficient as it introduces $nm$ non-parallelizable steps. This highlights the importance of the latent node and edge representations, which enable us to model the dependencies in the output efficiently because predicting all $\boldsymbol{I}^*_{i,j}$ can now happen in parallel. Furthermore, this changes our assumption on the incidence probabilities from that of independence to *conditional* independence on $\boldsymbol{E}_i$ and $\boldsymbol{V}_j$, and we apply a binary cross-entropy loss on each $\boldsymbol{I}_{i,j}$.

The binary classification over $\boldsymbol{I}^*_{i,j}$ highlights yet another reason for picking the incidence representation over the adjacency. Let us assume we are trying to learn a binary classifier that predicts for every

entry in the adjacency tensor whether it is 0 or 1. Removing all the 0 values (non-existing edges) from the training set will obviously not work out in the adjacency case. In contrast, an existing edge in the incidence matrix contains both 1s and 0s (except for the edge connecting all nodes), ensuring that a binary incidence classifier sees both positive and negative examples. However, an adjacency tensor has the advantage that the order of the entries is fully decided by the order of the nodes, which are given by the input $\boldsymbol{X}$ in our case. In the incidence matrix, the row order of the incidence matrix depends on the edges, which are orderless.

When comparing the predicted incidence matrix with a ground-truth matrix, we need to account for the orderless nature of the edges and the given order of the nodes. Hence, we require a loss function that is invariant toward reordering over the rows of the incidence matrix, but equivariant to reordering over the columns. We achieve this by matching every row in $\boldsymbol{I}$ with a row in the pruned ground-truth incidence matrix $\boldsymbol{I}^*$ (containing the existing edges), such that the binary cross-entropy loss $H$ over all entries is minimal:

$$\mathcal{L}(\boldsymbol{I}, \boldsymbol{I}) = \min_{\pi \in \Pi} \sum_{i,j} H(\boldsymbol{I}_{\pi(i),j}, \boldsymbol{I}^*_{i,j}) \tag{1}$$

Finding a permutation $\pi$ that minimizes the total loss is known as the linear assignment problem and we solve it with an efficient variant of the Hungarian algorithm [13, 14]. We discuss the implications on the computational complexity of this in Appendix B.

Having established the training objective in Equation 1, we can now offer a more formal argument on the soundness of supervising existing edges only while pruning the non-existing ones, where $\boldsymbol{J}$ can be understood as the *full* incidence matrix (proof in Appendix A):

**Proposition 1** (Supervising only existing edges). *Let $\boldsymbol{J} \in [\epsilon, 1)^{(2^n-1) \times n}$ be a matrix with at most $m$ rows for which $\exists j \colon \boldsymbol{J}_{ij} > \epsilon$, with a small $\epsilon > 0$. Similarly, let $\boldsymbol{J}^* \in \{0, 1\}^{(2^n-1) \times n}$ be a matrix with at most $m$ rows for which $\exists j \colon \boldsymbol{J}_{ij} = 1$. Let $\mathtt{prune}(\cdot)$ denote the function that maps from a $(2^n - 1) \times n$ matrix to a $k \times n$ matrix, by removing $(2^n - 1) - k$ rows where all values are $\leq \epsilon$. Then, for a constant $c = (2^n - 1 - k)n \cdot H(\epsilon, 0)$ and all such $\boldsymbol{J}$ and $\boldsymbol{J}^*$:*

$$\mathcal{L}(\boldsymbol{J}, \boldsymbol{J}^*) = \mathcal{L}(\mathtt{prune}(\boldsymbol{J}), \mathtt{prune}(\boldsymbol{J}^*)) + c \tag{2}$$

The matrix $\mathtt{prune}(\boldsymbol{J})$ is equivalent to the pruned incidence matrix that we defined earlier and $\mathtt{prune}(\boldsymbol{J}^*)$ is the corresponding pruned ground-truth. In practice, the $\epsilon$ corresponds to a lower bound on the $\log$ function in the entropy computation, for example, the lower bound in PyTorch [15] is -100. Since the losses in Equation 2 are equivalent up to an additive constant, the gradients of the parameters are exactly equal in both the pruned and non-pruned cases. Thus, pruning the non-existing edges does not affect learning, while significantly reducing the asymptotic memory requirements.

**Summary.** In set-to-hypergraph tasks, the number of different edges that can be predicted grows exponentially with the input set size. We address this computational limitation by representing the edge connections with the incidence matrix and pruning all non-existing edges *before* explicitly deciding for each edge whether it exists or not. We show that pruning the edges is sound when the loss function respects the permutation symmetry in the edges.

## 3   Scaling by skipping the non-essential gradients

In this section, we consider how to *learn* a model that can predict the pruned incidence matrix for arbitrarily complex problem instances. Some tasks may require more compute than others, resulting in worse performance or intractable models if not properly addressed. A naive approach would either increase the number of hidden dimensions or the depth of the neural network, which would further increase the memory requirement of backprop and quickly approach the memory limits of GPUs. Instead, we would like to increase the amount of sequential computation by reusing parameters. That is, we want the model $f$ to be recurrent, $\mathcal{H}^{t+1} = f(\boldsymbol{X}, \mathcal{H}^t)$ with $t$ denoting the $t$-th output of $f$ starting from a random initialization at $t = 0$, and $\mathcal{H}^t = (\boldsymbol{V}^t, \boldsymbol{E}^t, \boldsymbol{I}^t)$ represents the current prediction of the hypergraph at step $t$. Recurrent models are commonly applied to sequential data [16], where the input varies for each time step $t$, for example, the words in a sentence. In our case, we use the same input $\boldsymbol{X}$ at every step. Using a recurrent model, we can increase the total number of iterations — this scales the number of sequential computation steps — without increasing the number of parameters. However, the recurrence does not address the growing memory requirements of backprop, since the activations of each iteration still need to be kept in memory.

**Iterative refinement.** In the rest of this section, we present an efficient training algorithm that can scale to any number of iterations at a constant memory cost. We build on the idea that if each iteration applies a small refinement, then it becomes unnecessary to backprop through every iteration. We can define an iterative refinement as reducing the loss (by a little) after every iteration, $\mathcal{L}(\boldsymbol{I}^t, \boldsymbol{I}^*) < \mathcal{L}(\boldsymbol{I}^{t-1}, \boldsymbol{I}^*)$. Thus, the long-term dependencies between $\mathcal{H}^t$ for $t$'s that are far apart can also be ignored, since $f$ only needs to improve the current $\mathcal{H}^t$. We can encourage $f$ to iteratively refine the prediction $\mathcal{H}^t$, by applying the loss $\mathcal{L}$ after each iteration. This has the effect that $f$ learns to move the $\mathcal{H}^t$ in the direction of the negative gradient of $\mathcal{L}$, akin to a gradient descent update.

**Backprop with skips.** Similar to previous works that encourage iterative refinement through (indirect) supervision on the intermediate steps [17], we expect the changes of each step to be small. Thus, it stands to reason that supervising *every* step is unnecessary and redundant. This leads us to a more efficient training algorithm that skips iterations in the backward-pass of backprop. Algorithm 1 describes the training procedure in pseudocode (in a syntax similar to PyTorch [15]). The argument $S$ is a list of integers of length $N$ that is the number of gradient updates per mini-batch. Each gradient update consists of two phases, first $s \in S$ iterations without gradient accumulation and then $B$ iterations that add up the losses for backprop. Through these hyperparameters, we control the amount of resources used during training. Increasing hyperparameter $B$ allows for models that do not strictly decrease the loss after every step and require supervision over multiple subsequent steps. Note that having the input $\boldsymbol{X}$ at every refinement step is important so that the model does not forget the initial problem.

---

**Algorithm 1:** Backprop with skips

---
**Input:** $\boldsymbol{X}, \boldsymbol{I}^*, S, B$
$\mathcal{H} \leftarrow \texttt{initialize}(\boldsymbol{X})$
**for** $s$ **in** $S$ **:**
    **with** $\texttt{no\_grad}()$ **:**
        **for** $t$ **in** $\texttt{range}(s)$ **:**
            $\mathcal{H} \leftarrow f(\boldsymbol{X}, \mathcal{H})$
    $l \leftarrow 0$
    **for** $t$ **in** $\texttt{range}(B)$ **:**
        $\mathcal{H} \leftarrow f(\boldsymbol{X}, \mathcal{H})$
        $l \leftarrow l + \mathcal{L}(\mathcal{H}, \boldsymbol{I}^*)$
    $\texttt{backward}(l)$
    $\texttt{gradient\_step\_and\_reset}()$

---

**Summary.** Motivated by the need for more compute to address complex problems, we propose a method that increases the amount of sequential compute of the neural network without increasing the memory requirement at training time. Our training algorithm requires the model $f$ to perform iterative refining of the hypergraph, for which we present a method in the next section.

## 4 Scaling the set-to-hypergraph prediction model

In Section 2 and Section 3 we proposed two methods to overcome the memory scaling problems that appear for set-to-hypergraph tasks. To put these into practice, we need to specify a neural network $f$ that can produce an incidence matrix while preserving its symmetries. In what follows, we propose a recurrent neural network that iteratively refines latent node and edge features based on which it predicts the incidence matrix.

**Initialization.** As the starting point for the iterative refinement, we initialize the latent node features $\boldsymbol{V}^0$ from the input node features as $\boldsymbol{V}_i^0 = \boldsymbol{W}\boldsymbol{X}_i + \boldsymbol{b}$, where $\boldsymbol{W} \in \mathbb{R}^{d_{\boldsymbol{V}} \times d_{\boldsymbol{X}}}, \boldsymbol{b} \in \mathbb{R}^{d_{\boldsymbol{V}}}$ are learnable parameters. The affine map allows for hidden dimensions $d_{\boldsymbol{V}}$ that is different from the input feature dimensions $d_{\boldsymbol{X}}$. An informed initialization similar to the nodes is not available for the edges and the incidence matrix, since these are what we aim to predict. Instead, we choose an initialization scheme that respects the permutation symmetry of a set of edges while also ensuring that each edge starts out differently. The last point is necessary for permutation-equivariant operations to distinguish between different edges [18]. The random initialization $\boldsymbol{E}_i^0 \sim \mathcal{N}(\boldsymbol{\mu}, \text{diag}(\boldsymbol{\sigma}))$, with shared learnable parameters $\boldsymbol{\mu} \in \mathbb{R}^{d_{\boldsymbol{E}}}$ and $\boldsymbol{\sigma} \in \mathbb{R}^{d_{\boldsymbol{E}}}$ fulfills both these properties, as it is highly unlikely for two samples to be identical.

**Conditional independence.** We want the incidence probabilities $\boldsymbol{I}_{i,j}$ to be conditionally independent of each other given $\boldsymbol{E}_i$ and $\boldsymbol{V}_j$. A straightforward way to model this is by concatenating both vectors (denoted with $[\cdot]$) and applying an MLP with a sigmoid activation on the scalar output:

$$\boldsymbol{I}_{i,j}^t = \texttt{MLP}\left(\left[\boldsymbol{E}_i^{t-1}, \boldsymbol{V}_j^{t-1}\right]\right) \tag{3}$$

The superscripts point out that we produce a new incidence matrix for step $t$ based on the edge and node vectors from the previous step. Note that we did not specify an initialization for the incidence matrix, since we directly replace it in the first step.

**Iterative refinement.** The training algorithm in Section 3 assumes that $f$ performs iterative refinement on $\mathcal{H}^t$, but leaves open the question on how to design such a model. Instead of iteratively refining the incidence matrix, i.e., the only term that appears in the loss (Equation 1), we focus on refining the edges and nodes.

A refinement step for the latent feature of some node $V_i$ needs to account for the rest of the hypergraph, which also changes with each iteration. For this purpose, we apply the permutation-equivariant DeepSets [19] to produce node updates dependent on the full set of nodes from the previous iteration $V^{t-1}$. The permutation-equivariance of DeepSets implies that the output set retains the input order; thus it is sensible to refer to $V_i^t$ as the updated version of the *same* node $V_i^{t-1}$ from the previous step. Furthermore, we concatenate the aggregated neighboring edges weighted by the incidence probabilities $\rho_{E \to V}(j,t) = \sum_{i=1}^{k} I_{i,j}^t E_i^{t-1}$, to incorporate the relational structure between the nodes. This aggregation works akin to message passing in graph neural networks [20]. To ensure that the model does not forget the initial problem, as represented by $X$, we include a skip connection from the input to every refinement step. This is an indispensable aspect of the architecture, without which backprop with skips would not work. Instead of directly concatenating the raw features $X_i$, we use the initial nodes $V_i^0$. Finally, we express the refinement part for the nodes as:

$$V^t = \texttt{DeepSets}\left(\left\{\left[V_j^{t-1}, \rho_{E \to V}(j,t), V_j^0\right] \middle| j = 1 \ldots n\right\}\right) \tag{4}$$

The updates to the edges $E^t$ mirror that of the nodes, except for the injection of the input set. Together with the aggregation function $\rho_{V \to E}(i,t) = \sum_{j=1}^{n} I_{i,j}^t V_j^{t-1}$, we can update the edges as:

$$E^t = \texttt{DeepSets}\left(\left\{\left[V_i^{t-1}, \rho_{V \to E}(i,t)\right] \middle| i = 1 \ldots k\right\}\right) \tag{5}$$

By sharing the parameters between different refinement steps, we naturally obtain a recurrent model. Previous works on recurrent models [21] saw improvements in training convergence by including layer normalization [22] between each iteration. Shortcut connections in ResNets [23] have been shown to encourage iterative refinement of the latent vector [17]. We add both shortcut connections and layer normalization to the updates in Equation 4 and Equation 5. Although we prune the negative edges, we still want to predict a variable number thereof. To achieve that we simply extend the incidence model in Equation 3 with an existence indicator:

$$\hat{I}_i^t = \sigma_i^t I_i^t \tag{6}$$

This can be seen as factorizing the probability into "$p(E_i$ incident to $V_j) \cdot p(E_i$ exists)" and replaces the aggregation weights in $\rho_{E \to V}$ and $\rho_{V \to E}$.

**Summary.** We propose a model that fulfills the requirements of our scalable set-to-hypergraph training framework from Section 2 and Section 3. By adding shortcut connections, we encourage it to perform iterative refinements on the hypergraph while being permutation equivariant with respect to both the nodes and the edges.

## 5 Experiments

In this section, we evaluate our approach on multiple set-to-hypergraph tasks, in order to compare to prior work and examine the main design choices. We refer to Appendix D for further details, results, and ablations. The code for reproducing the experiments is available at `https://github.com/davzha/recurrently_predicting_hypergraphs`.

### 5.1 Scaling set-to-hypergraph prediction

First, we compare our model from Section 4 on three different set-to-hypergraph tasks against the state-of-the-art. The comparison offers insight into the differences between predicting the incidence matrix and predicting the adjacency tensors.

**Table 1: Particle partitioning results.** On three jet types performance was measured in F1 score and adjusted rand index (ARI) for 11 different seeds. Our method outperforms the baselines on bottom and charm jets while being competitive on light jets.

| | bottom jets | | charm jets | | light jets | |
|---|---|---|---|---|---|---|
| Model | F1 | ARI | F1 | ARI | F1 | ARI |
| Set2Graph | $0.646_{\pm 0.003}$ | $0.491_{\pm 0.006}$ | $0.747_{\pm 0.001}$ | $0.457_{\pm 0.004}$ | $0.972_{\pm 0.001}$ | $0.931_{\pm 0.003}$ |
| Set Transformer | $0.630_{\pm 0.004}$ | $0.464_{\pm 0.007}$ | $0.747_{\pm 0.003}$ | $0.466_{\pm 0.007}$ | $0.970_{\pm 0.001}$ | $0.922_{\pm 0.003}$ |
| Slot Attention | $0.600_{\pm 0.012}$ | $0.411_{\pm 0.021}$ | $0.728_{\pm 0.008}$ | $0.429_{\pm 0.016}$ | $0.963_{\pm 0.002}$ | $0.895_{\pm 0.009}$ |
| Ours | $0.679_{\pm 0.002}$ | $0.548_{\pm 0.003}$ | $0.763_{\pm 0.001}$ | $0.499_{\pm 0.002}$ | $0.972_{\pm 0.001}$ | $0.926_{\pm 0.002}$ |

**Baselines.** Our main comparison is against Set2Graph [2], which is a strong and representative baseline for approaches that predict the adjacency structure, which we generally refer to as adjacency-based approaches. Serviansky et al. [2] modify the task in two of the benchmarks, to avoid storing an intractably large adjacency tensor. We explain in Appendix D how this affects the comparison. Additionally, we compare to Set Transformer [24] and Slot Attention [21], which we adapt to the set-to-hypergraph setting by treating the output as the pruned set of edges and producing the incidence matrix with the MLP from Equation 3. We refer to these two as incidence-based approaches that also include our model.

**Particle partitioning.** Particle colliders are an important tool for studying the fundamental particles of nature and their interactions. During a collision, several particles are emanated and measured by nearby detectors, while some particles decay beforehand. Identifying which measured particles share a common progenitor is an important subtask in the context of vertex reconstruction [25]. We can treat this as a set-to-hypergraph task: the set of measured particles is the input set and the common progenitors are the edges of the hypergraph. We use a simulated dataset of 0.9M data-sample with the default train/validation/test split [2, 25]. Each data-sample is generated from on one of three different distributions for which we report the results separately: *bottom jets*, *charm jets* and *light jets*. The ground-truth target is the incidence matrix that can also be interpreted as a partitioning of the input elements since every particle has exactly one progenitor (edge).

In Table 1 we report the performances on each type of jet as the F1 score and Adjusted Rand Index (ARI). Our method outperforms all alternatives on bottom and charm jets while being competitive on light jets.

**Convex hull.** The convex hull of a finite set of $d$-dimensional points can be efficiently represented as the set of simplices that enclose all points. In the 3D case, each simplex consists of 3 points that together form a triangle. For the general $d$-dimensional case, the valid incidence matrices are limited to those with $d$ incident vertices per edge. Finding the convex hull is an important and well-understood task in computational geometry, with optimal exact solutions [11, 26]. Nonetheless, predicting the convex hull for a given set of points poses a challenging problem for current machine learning methods, especially when the number of points increases [2, 5]. We generate an input set by drawing $n$ 3-dimensional vectors from one of two distributions: Gaussian or spherical. For the Gaussian setting, points are sampled i.i.d. from a standard normal distribution. For the spherical setting, we additionally normalize each point to lie on the unit sphere. Following Serviansky et al. [2], we use $n{=}30$, $n{=}50$ and $n{\in}[20\,..\,100]$, where in the latter case the input set size varies between 20 and 100.

Table 2 shows our results. Our method consistently outperforms all the baselines by a considerable margin. In contrast to Set2Graph, our model does not suffer from a drastic performance decline when increasing the input set size from 30 to 50. Furthermore, based on the results in the Gaussian setting, we also observe that all the incidence-based approaches handle varying input sizes much better than the adjacency-based approach.

**Delaunay triangulation.** A Delaunay triangulation of a finite set of 2D points is a set of triangles for which the circumcircles of all triangles have no point lying inside. When there exists more than one such set, Delaunay triangulation aims to maximize the minimum angle of all triangles. We consider the same learning task and setup as Serviansky et al. [2], who frame Delaunay triangulation as a mapping from a set of 2D points to the set of Delaunay edges, represented by the adjacency

**Table 2: Convex hull results** measured as F1 score. Our method outperforms all baselines considerably for all settings and set sizes ($n$).

| Model | Spherical | | | Gaussian | | |
|---|---|---|---|---|---|---|
| | $n{=}30$ | $n{=}50$ | $n{\in}[20\mathinner{.\,.}100]$ | $n{=}30$ | $n{=}50$ | $n{\in}[20\mathinner{.\,.}100]$ |
| Set2Graph | 0.780 | 0.686 | 0.535 | 0.707 | 0.661 | 0.552 |
| Set Transformer | 0.773 | 0.752 | 0.703 | 0.741 | 0.727 | 0.686 |
| Slot Attention | 0.668 | 0.629 | 0.495 | 0.662 | 0.665 | 0.620 |
| Ours | 0.892 | 0.868 | 0.823 | 0.851 | 0.831 | 0.809 |

**Table 3: Delaunay triangulation results** for different set sizes ($n$). Our method outperforms Set2Graph on all metrics.

| Model | $n{=}50$ | | | | $n{\in}[20\mathinner{.\,.}80]$ | | | |
|---|---|---|---|---|---|---|---|---|
| | Acc | Pre | Rec | F1 | Acc | Pre | Rec | F1 |
| Set2Graph | 0.984 | 0.927 | 0.926 | 0.926 | 0.947 | 0.736 | 0.934 | 0.799 |
| Ours | 0.989 | 0.953 | 0.946 | 0.950 | 0.987 | 0.945 | 0.942 | 0.943 |

matrix. Since this is essentially a set-to-*graph* problem instead of a set-to-hypergraph one, we adapt our method for efficiency reasons, as we describe in Appendix D. We generate the input sets of size $n$, by sampling 2-dimensional vectors uniformly from the unit square, with $n{=}50$ or $n \in [20\mathinner{.\,.}80]$.

In Table 3, we report the results for Set2Graph [2] and our adapted method. Since the other baselines were not competitive in convex hull finding, we do not repeat them here. Our method again outperforms Set2Graph on all metrics.

**Summary.**  By predicting the positive edges only, we can significantly reduce the amount of required memory for set-to-hypergraph tasks. On three different benchmarks, we observe performance improvements when using this incidence-based approach, compared to the adjacency-based baseline. Interestingly, our method does *not* see a large discrepancy in performance between different input set sizes, both in convex hull finding and Delaunay triangulation. We attribute this to the recurrence of our iterative refinement scheme, which we look into next.

## 5.2  Ablations

**Effects of increasing (time) complexity.**  The *intrinsic* complexity of finding a convex hull for a $d$-dimensional set of $n$ points is in $\mathcal{O}(n \log(n) + n^{\lfloor \frac{d}{2} \rfloor})$ [11]. This scaling behavior offers an interesting opportunity to study the effects of increasing (time) complexity on model performance. The time complexity implies that *any* algorithm for convex hull finding scales super-linearly with the input set size. Since our learned model is not considered an algorithm that (exactly) solves the convex hull problem, the implications become less clear. In order to assess the relevancy of the problem's complexity for our approach, we examine the relation between the number of refining steps and increases in the intrinsic resource requirement. The following experiments are all performed with standard backprop, in order to not introduce additional hyperparameters that may affect the conclusions.

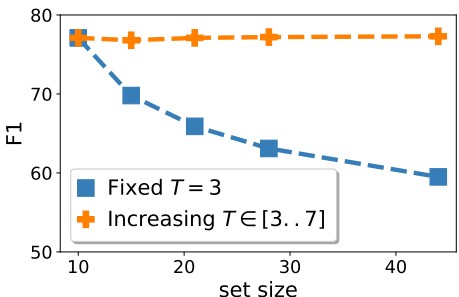

**Figure 2: Increasing complexity.** Increasing the iterations counteracts the performance decline from larger set sizes.

First, we examine the performance of our approach with 3 iterations, trained on increasing set sizes $n{\in}[10\mathinner{.\,.}50]$. In Figure 2 we observe a monotone drop in performance when training with the same number of iterations. The negative correlation between the set size and the performance confirms

a relationship between the computational complexity and the difficulty of the learning task. Next, we examine the performance for varying numbers of iterations and set sizes. We refer to the setting, where the number of iterations is 3 and set size $n{=}10$, as the base case. All other set sizes and number of iterations are chosen such that the performance matches the base case as closely as possible. In Figure 2, we observe that the required number of iterations increases with the input set size, further highlighting that an increase in the number of iterations actually suffices in counteracting the performance decrease. Furthermore, we observe that the number of refinement steps scales sub-linearly with the set size, different from what we would expect based on the complexity of the problem. We speculate this is due to the parallelization of our edge finding process, differing from incremental approaches that produce one edge at a time.

**Efficiency of backprop with skips.**   To assess the efficiency of backprop with skips, we compare to truncated backpropagation through time (TBPTT) [27]. We consider two variants of our training algorithm: 1. Skipping iterations at fixed time steps and 2. Skipping randomly sampled time steps. In both the fixed and random skips versions, we skip half of the total iterations. We train all models on convex hull finding in 3-dimensions for 30 spherically distributed points. In addition, we include baselines trained with standard backprop that contingently inform us about performance degradation incurred by our method or TBPTT. Standard backprop increases the memory footprint linearly with the number of iterations $T$, inevitably exceeding the available memory at some threshold. Hence, we deliberately choose a small set size in order to accommodate training with backprop for $T{\in}\{4, 16, 32\}$ number of iterations. We illustrate the differences between standard backprop, TBPTT and our backprop with skips in Figure 5 in the Appendix.

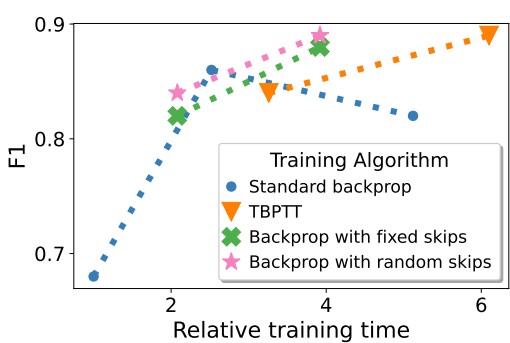

**Figure 3: Training time of backprop with skips.** Relative training time and performance for different $T{\in}\{4, 16, 32\}$. All runs require the same memory, except standard backprop $T{\in}\{16, 32\}$, which require more.

The results in Figure 3 demonstrate that skipping half of the iterations in the backward-pass significantly decreases the training time without hurting predictive performance. When the memory budget is constricted to 4 iterations in the backward-pass, both TBPTT and backprop with skips outperform standard backprop considerably. We provide a detailed discussion of the computational complexity of our framework in Appendix B.

**Recurrent vs. stacked.**   Recurrence plays a crucial role in enabling more computation without an increase in the number of parameters. By training the recurrent model using backprop with skips, we can further reduce the memory cost during training to a constant amount. Since our proposed training algorithm from Section 3 encourages iterative refinement akin to gradient descent, it is natural to believe that the weight-tying aspect of recurrence is a good inductive bias for modeling this. A reason for thinking so is that the "gradient" should be the same for the same $\boldsymbol{I}$, no matter at which iteration it is computed. Hence, we compare the recurrent model against a non-weight-tied (stacked) version that applies different parameters at each iteration.

First, we compare the models trained for 3 to 9 refinement steps. In Figure 4a, we show that both cases benefit from increasing the iterations. Adding more iterations beyond 6 only slightly improves the performance of the stacked model, while the recurrent version still benefits, leading to an absolute difference of 0.03 in F1 score for 9 iterations.

Next, we train both versions with 15 iterations until they achieve a similar validation performance, by stopping training early on the recurrent model. The results in Figure 4b show that the recurrent variant performs better when tested at larger set sizes than trained, indicating an improved generalization ability.

**Learning higher-order edges.**   The particle partitioning experiment exemplifies a case where a single edge can connect up to 14 vertices. Set2Graph [2] demonstrates that in this specific case

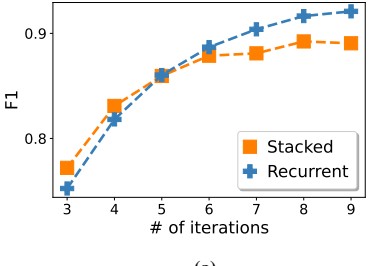 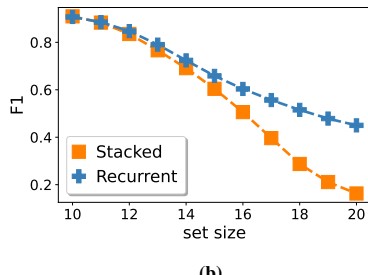

(a)  (b)

**Figure 4: Recurrent vs. stacked.** (a) Performance for different numbers of iterations. (b) Extrapolation performance on $n \in [11 \mathinner{.\,.} 20]$ for models trained with set size $n{=}10$. We stop training the recurrent model early, to match the validation performance of the stacked on $n{=}10$. The recurrent model derives greater benefits from adding iterations and generalizes better.

it is possible to approximate the hypergraph with a graph. They leverage the fact that any vertex is incident to exactly one edge and apply a post-processing step that constructs the edges from noisy cliques. Instead, we consider a task for which no straightforward graph-based approximation exists. Specifically, we consider convex hull finding in 10-dimensional space for 13 standard normal distributed points. We train with $T{=}32$ total iterations and $B{=}4$ backprop iterations. The test performance reaches an F1 score of 0.75, clearly demonstrating that the model managed to learn. This result demonstrates the improved scaling behavior can be leveraged for tasks that are computationally out of reach for adjacency-based approaches.

## 6   Related Work

Set2Graph [2] is a family of maximally expressive permutation equivariant neural networks that map from an input set to (hyper)graphs. They show that their method outperforms many popular alternatives, including Siamese networks [28] and graph neural networks [29] applied to a $k$-NN induced graph. [2] extend the general idea, of applying a scalar-valued adjacency indicator function on all pairs of nodes [30], to the $l$-edge case (edges that connect $l$ nodes). In Set2Graph, for each $l$ the adjacency structure is modeled by an $l$-tensor, requiring memory in $\mathcal{O}(n^l)$. This becomes intractable already for small $l$ and moderate set sizes. By pruning the negative edges, our approach scales in $\mathcal{O}(nk)$, making it applicable even when $l{=}n$. Recent works on set prediction map a learned initial set [24, 31] or a randomly initialized set [21, 32, 33] to the target space. Out of these, the closest one to our hypergraph refining approach is Slot Attention [21], which recurrently applies the Sinkhorn operator [34] in order to associate each element in the input set with a single slot (hyperedge). None of the prior works on set prediction consider the set-to-hypergraph task, but some can be naturally extended by mapping the input set to the set of positive edges, an approach similar to ours.

## 7   Conclusion and future work

By representing and supervising the set of positive edges only, we substantially improve the asymptotic scaling and enable learning tasks with higher-order edges. On common benchmarks, we have demonstrated that our method outperforms previous works while offering a more favorable asymptotic scaling behavior. In further evaluations, we have highlighted the importance of recurrence for addressing the intrinsic complexity of problems. We identify the Hungarian matching [13] as the main computational bottleneck during training. Replacing the Hungarian matched loss with a faster alternative, like a learned energy function [33], would greatly speed up training for tasks with a large maximum number of edges. Our empirical analysis is limited to datasets with low dimensional inputs. Learning on higher dimensional input data might require extensions to the model that can make larger changes to the latent hypergraph than is feasible with small iterative refinement steps. The idea here is similar to the observation from Jastrzebski et al. [17] for ResNets [23] that also encourage iterative refinement: earlier residual blocks apply large changes to the intermediate features while later layers perform (small) iterative refinements.

## Acknowledgement

This work is part of the research programme Perspectief EDL with project number P16-25 project 3, financed by the Dutch Research Council (NWO) domain Applied and Engineering Sciences (TTW).

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

# A    Proof: Supervising positive edges only suffices

**Proposition 1** (Supervising only existing edges). *Let $J \in [\epsilon, 1)^{(2^n-1) \times n}$ be a matrix with at most $m$ rows for which $\exists j \colon J_{ij} > \epsilon$, with a small $\epsilon > 0$. Similarly, let $J^* \in \{0,1\}^{(2^n-1) \times n}$ be a matrix with at most $m$ rows for which $\exists j \colon J_{ij} = 1$. Let $\mathtt{prune}(\cdot)$ denote the function that maps from a $(2^n - 1) \times n$ matrix to a $k \times n$ matrix, by removing $(2^n - 1) - k$ rows where all values are $\leq \epsilon$. Then, for a constant $c = (2^n - 1 - k)n \cdot H(\epsilon, 0)$ and all such $J$ and $J^*$:*

$$\mathcal{L}(\boldsymbol{J}, \boldsymbol{J}^*) = \mathcal{L}(\mathtt{prune}(\boldsymbol{J}), \mathtt{prune}(\boldsymbol{J}^*)) + c \tag{2}$$

*Proof.* We shorten the notation with $\boldsymbol{I} = \mathtt{prune}(\boldsymbol{J})$ and $\boldsymbol{I}^* = \mathtt{prune}(\boldsymbol{J})^*$, making the relation to the incidence matrix $\boldsymbol{I}$ defined in Section 2 explicit. Since $\mathcal{L}$ is invariant to permutations over the rows of its input matrices, we can assume w.l.o.g. that the not-pruned rows are the first $k$ rows, $\boldsymbol{J}_{:k} = \boldsymbol{I}$ and $\boldsymbol{J}^*_{:k} = \boldsymbol{I}^*$. For improved readability, let $\hat{H}(\boldsymbol{J}_{\pi(i)}, \boldsymbol{J}^*_i) = \sum_j H(\boldsymbol{J}_{\pi(i),j}, \boldsymbol{J}^*_{i,j})$ denote the element-wise binary cross-entropy, thus the loss in Equation 1 can be rewritten as $\mathcal{L}(\boldsymbol{J}, \boldsymbol{J}^*) = \min_{\pi \in \Pi} \sum_i \hat{H}(\boldsymbol{J}_{\pi(i)}, \boldsymbol{J}^*_i)$.

First, we show that there exists an optimal assignment between $\boldsymbol{J}, \boldsymbol{J}^*$ that assigns the first $k$ rows equally to an optimal assignment between $\boldsymbol{I}, \boldsymbol{I}^*$. More formally, for an optimal assignment $\pi_I \in \arg\min_{\pi \in \Pi} \sum_i \hat{H}(\boldsymbol{I}_{\pi(i)}, \boldsymbol{I}^*_i)$ we show that there exists an optimal assignment $\pi_J \in \arg\min_{\pi \in \Pi} \sum_i \hat{H}(\boldsymbol{J}_{\pi(i)}, \boldsymbol{J}^*_i)$ such that $\forall 1 \leq i \leq k \colon \pi_J(i) = \pi_I(i)$. If $\pi_J(i) \leq k$ for all $1 \leq i \leq k$ then the assignment for the first $k$ rows is also optimal for $\boldsymbol{I}, \boldsymbol{I}^*$. So we only need to show that there exists a $\pi_J$ such that $\pi_J(i) \leq k$ for all $1 \leq i \leq k$. Let $\pi_J$ be an optimal assignment that maps an $i < k$ to $\pi_J > k$. Since $\pi_J$ is a bijection, there also exists a $j < k$ that $\pi_J^{-1}(j) > k$ assigns to. The corresponding loss terms are lower bounded as follows:

$$\hat{H}(\boldsymbol{J}_i, \boldsymbol{J}^*_{\pi_J(i)}) + \hat{H}(\boldsymbol{J}_{\pi_J^{-1}(j)}, \boldsymbol{J}^*_j) \tag{7}$$

$$= \hat{H}(\boldsymbol{J}_i, \boldsymbol{0}) + \hat{H}(\boldsymbol{\epsilon}, \boldsymbol{J}^*_j) \tag{8}$$

$$= -\sum_{l=1}^n \log(1 - \boldsymbol{J}_{i,l}) + \boldsymbol{J}^*_{j,l} \log(\epsilon) + (1 - \boldsymbol{J}^*_{j,l}) \log(1 - \epsilon) \tag{9}$$

$$\geq -\sum_{l=1}^n (1 - \boldsymbol{J}^*_{j,l}) \log(1 - \boldsymbol{J}_{i,l}) + \boldsymbol{J}^*_{j,l} \log(\epsilon) + (1 - \boldsymbol{J}^*_{j,l}) \log(1 - \epsilon) \tag{10}$$

$$\geq -\sum_{l=1}^n (1 - \boldsymbol{J}^*_{j,l}) \log(1 - \boldsymbol{J}_{i,l}) + \boldsymbol{J}^*_{j,l} \log(\boldsymbol{J}_{i,l}) + (1 - \boldsymbol{J}^*_{j,l}) \log(1 - \epsilon) \tag{11}$$

$$= \hat{H}(\boldsymbol{J}_i, \boldsymbol{J}^*_j) - \sum_{l=1}^n (1 - \boldsymbol{J}^*_{j,l}) \log(1 - \epsilon) \tag{12}$$

$$\geq \hat{H}(\boldsymbol{J}_i, \boldsymbol{J}^*_j) - \sum_{l=1}^n \log(1 - \epsilon) \tag{13}$$

$$= \hat{H}(\boldsymbol{J}_i, \boldsymbol{J}^*_j) + \hat{H}(\boldsymbol{\epsilon}, \boldsymbol{0}) \tag{14}$$

Equality of Equation 8 holds since all rows with index $> k$ are $\epsilon$-vectors in $\boldsymbol{J}$ and zero-vectors in $\boldsymbol{J}^*$. The inequality in Equation 11 holds since all values in $\boldsymbol{J}$ are lower bounded by $\epsilon$. Thus, we have shown that either there exists no optimal assignment $\pi_J$ that maps from a value $\leq k$ to a value $> k$ (which is the case when $\hat{H}(\boldsymbol{J}_i, \boldsymbol{J}^*_{\pi_J(i)}) + \hat{H}(\boldsymbol{J}_{\pi_J^{-1}(j)} > \hat{H}(\boldsymbol{J}_i, \boldsymbol{J}^*_j) + \hat{H}(\boldsymbol{\epsilon}, \boldsymbol{0}))$ or that there exists an equally good assignment that does not mix between the rows below and above $k$. Since the pruned rows are all identical, any assignment between these result in the same value $(2^n - 1 - k)\hat{H}(\boldsymbol{\epsilon}, \boldsymbol{0}) = (2^n - 1 - k)n \cdot H(\epsilon, 0)$ that only depends on the number of pruned rows $2^n - 1 - k$ and number of columns $n$. ☐

# B    Computational complexity

The space complexity of the hypergraph representation presented in Section 2 is in $\mathcal{O}(nm)$, offering an efficient representation for hypergraphs when the maximal number of edges $m$ is low, relative to

the number of all possible edges $m \ll 2^n$. Problems that involve edges connecting many vertices benefit from this choice of representation, as the memory requirement is independent of the maximal connectivity of an edge. This differs from the adjacency-based approaches that not only depend on the maximum number of nodes an edge connects, but scale exponentially with it. In practice, this improvement from $\mathcal{O}(2^n)$ to $\mathcal{O}(mn)$ is important even for moderate set sizes because the amount of required memory determines whether it is possible to use efficient hardware like GPUs. We showcase this in Section D.4.

Backprop with skips, introduced Section 3, further scales the memory requirement by a factor of $B$ that is the number of iterations to backprop through in a single gradient update step. Notably, this scales constantly in the number of gradient update steps $N$ and iterations skipped during backprop $\sum_i S_i$. Hence, we can increase the number of recurrent steps to adapt the model to the problem complexity (which is important, as we show in Section 5.2), at a constant memory footprint.

To compute the loss in Equation 1, we apply a modified Jonker-Volgenant algorithm [14, 35, 36] that finds the minimum assignment between the rows of the predicted and the ground truth incidence matrices in $\mathcal{O}(m^3)$. In practice, this can be the main bottleneck of the proposed method when the number of edges becomes large. For problems with $m \ll n$, the runtime complexity is especially efficient since it is independent of the number of nodes.

**Comparison to Set2Graph.** When we compare Set2Graph and our method for hyperedges that connect two nodes (i.e., a normal graph) then Set2Graph has an efficiency advantage. For example, the authors report for Set2Graph a training time of 1 hour (on a Tesla V100 GPU) for Delaunay triangulations, while our method takes up to 9 hours (on a GTX 1080 Ti GPU). Nevertheless, the major bottleneck of the Set2Graph method is its memory and time complexity, which in practice can lead to intractable memory requirements even for small problems. For example, the convex hull problem in 10-dimensional space over 13 points requires more than 500GB ($\approx 4 * 13^10$) just for storing the adjacency tensor – that is without even considering the intermediate neural network activations. Even when we keep the space 3-dimensional as in the experiments reported by Set2Graph, it already struggles with memory requirements as the authors point out themselves. They circumvent this issue by considering an easier local version of the problem and restrict the adjacent nodes to the k-Nearest-Neighbors with k=10. In conclusion, when the hypergraph has relatively few edges, then our framework offers a much better scaling than Set2Graph. In practice, this does not merely translate to faster runtime but turns tasks that were previously intractable into feasible tasks.

## C    Backprop with skips

Our backprop with skips training consists of two aspects:

- Truncation of the backprop through time, and
- Skipping the gradient calculations for some intermediate $f(\boldsymbol{X}, \boldsymbol{\mathcal{H}})$ as specified by the $S$.

To understand what the effect of truncation is, we first consider the case of standard backprop. When training a residual neural network with standard backprop it is possible to expand the loss function around $\boldsymbol{\mathcal{H}}^t$ from previous iterations $t < T$ as $\mathcal{L}(\boldsymbol{\mathcal{H}}^T) = \mathcal{L}(\boldsymbol{\mathcal{H}}^t) + \sum_{i=t}^{T-1} f(\boldsymbol{X}, \boldsymbol{\mathcal{H}}^i)\frac{\partial \mathcal{L}(\boldsymbol{\mathcal{H}}^i)}{\partial \boldsymbol{\mathcal{H}}^i} + \mathcal{O}(f^2(\boldsymbol{X}, \boldsymbol{\mathcal{H}}^i)$ (see equation 4 in [13]). Jastrzebski et al. [13] point out that the sum terms' gradients point into the same half space as that of $\frac{\partial \mathcal{L}(\boldsymbol{\mathcal{H}}^i)}{\partial \boldsymbol{\mathcal{H}}^i}$, implying that $\sum_{i=t}^{T-1} \mathcal{L}(\boldsymbol{\mathcal{H}}^i)$ is also a descent direction for $\mathcal{L}(\boldsymbol{\mathcal{H}}_T)$. Thus, truncation can be interpreted as removing $\mathcal{L}(\boldsymbol{\mathcal{H}}_i)$ terms from the loss function for earlier steps $i$. We remedy this, by explicitly adding (back) the $\mathcal{L}(\boldsymbol{\mathcal{H}}_i)$ terms to the training objective. The second aspect of backprop with skips involves skipping gradients of some intermediate steps $f(\boldsymbol{X}, \boldsymbol{\mathcal{H}}^i)$ entirely. Given a specific data point $\boldsymbol{X}$, we view $f$ as approximating the gradient $\frac{\partial \boldsymbol{\mathcal{H}}(t)}{\partial t}$ where we denote the time step $t$ as the argument of $H$ instead of as the subscript. From this viewpoint, we are learning a vector field (analog to neural ODE [37] or diffusion models [38]) over the space of $\boldsymbol{\mathcal{H}}$ which we train by randomly sampling different values for the $\boldsymbol{\mathcal{H}}^t$'s.

## D    Experimental details

In this section, we provide further details on the experimental setup and additional results.

## D.1 Particle partitioning

The problem considers the case where particles are collided at high energy, resulting in multiple particles shooting out from the collision. Each example in the dataset consists of the input set, which corresponds to the measured outgoing particles, and the ground truth partition of the input set. Each element in the partition is a subset of the input set and corresponds to some intermediate particle that was not measured, because it decayed into multiple particles before it could reach the sensors. The learning task consists of inferring which elements in the input set originated from the same intermediate particle. Note that the particle partitioning task bears resemblance to the classical clustering setting. It can be understood as a meta-learning clustering task, where both the number of clusters and the similarity function depend on the context that is given by $X$. That is why clustering algorithms such as $k$-means cannot be directly applied to this task. For more information on how this task fits into the area of particle physics more broadly, we refer to Shlomi et al. [1].

**Dataset.** We use the publicly available dataset of 0.9M data-sample with the default train/validation/test split [2, 25]. The input sets consist of 2 to 14 particles, with each particle represented by 10 features. The target partitioning indicate the common progenitors and restrict the valid incidence matrices to those with a single incident edge per node.

**Setup.** While Set2Graph is only one instance of an adjacency-based approach, [2] show that it outperforms many popular alternatives: Siamese networks, graph neural networks and a non-learnable geometric-based baseline. All adjacency-based approaches incur a prohibitively large memory cost when predicting edges with high connectivity. In the case of particle partitioning, Set2Graph resorts to only predicting edges with at most 2 connecting nodes, followed by an additional heuristic to infer the partitions [2]. In contrast to that, all the incidence-based approaches do not require the additional post-processing step at the end.

We simplify the hyperparameter search by choosing the same number of hidden dimensions $d$ for the latent vector representations of both the nodes $d_V$ and the edges $d_E$. In all runs dedicated to searching $d$, we set the number of total iterations $T=3$ and backpropagate through all iterations. We start with $d=32$ and double it, until an increase yields no substantial performance gains on the validation set, resulting in $d=128$. In our reported runs, we use $T=16$ total iterations, $B=4$ backprop iterations, $N=2$ gradient updates per mini-batch, and a maximum of 10 edges.

We apply the same $d=128$ to both the Slot Attention and Set Transformer baselines. Similar to the original version [21], we train Slot Attention with 3 iterations. Attempts with more than 3 iterations resulted in frequent divergences in the training losses. We attribute this behavior to the recurrent sinkhorn operation, that acts as a contraction map, forcing all slots to the same vector in the limit.

We train all models using the Adam optimizer [39] with a learning rate of 0.0003 for 400 epochs and retain the parameters corresponding to the lowest validation loss. All models additionally minimize a soft F1 score [2]. Since each particle can only be part of a single partition, we choose the one with the highest incidence probability at test time. Our model has 268162 trainable parameters, similar to 251906 for the Slot Attention baseline, but less than 517250 for Set Transformer and 461289 for Set2Graph [2]. The total training time is less than 12 hours on a single GTX 1080 Ti and 10 CPU cores.

The maximum number of edges is set to $m = 10$.

**Further results.** For completeness, we also report the results for the rand index (RI) in Table 4.

## D.2 Convex hull finding

On convex hull finding in 3D, we compare our method to the same baselines as on the particle partitioning task.

**Setup.** Set2Graph learns to map the set of 3D points to the 3rd order adjacency tensor. Since storing this tensor in memory is not feasible, they instead concentrate on a local version of the problem, which only considers the $k$-nearest neighbors for each point [2]. We train our method with $T_{\text{total}}=48$, $T_{\text{BPTT}}=4$, $N_{\text{BPTT}}=6$ and set $k$ equal to the highest number of triangles in the training data. At test time, a prediction admits an edge $e_i$ if its existence indicator $\sigma_i > 0.5$. Each edge is incident to the three nodes with the highest incidence probability. We apply the same hyperparameters, architectures

**Table 4: Additional particle partitioning results.** On three jet types performance measured as rand index (RI). Our method outperforms the baselines on bottom and charm jets, while being competitive on light jets.

|  | **bottom jets** | **charm jets** | **light jets** |
|---|---|---|---|
| Model | RI | RI | RI |
| Set2Graph | $0.736_{\pm 0.004}$ | $0.727_{\pm 0.003}$ | $0.970_{\pm 0.001}$ |
| Set Transformer | $0.734_{\pm 0.004}$ | $0.734_{\pm 0.004}$ | $0.967_{\pm 0.002}$ |
| Slot Attention | $0.703_{\pm 0.013}$ | $0.714_{\pm 0.009}$ | $0.958_{\pm 0.003}$ |
| Ours | $0.781_{\pm 0.002}$ | $0.751_{\pm 0.001}$ | $0.969_{\pm 0.001}$ |

and optimizer as in the particle partitioning experiment, except for: $T{=}48$, $B{=}4$, $N{=}6$. Since we do not change the model, the number of parameters remains at 268162 for our model. This notably differs to Set2Graph, which reports an increased parameter count of 1186689 [2]. We train our method until we observe no improvements on the F1 validation performance for 20 epochs, with a maximum of 1000 epochs. The set-to-set baselines are trained for 4000 epochs, and we retain the parameters resulting in the highest f1 score on the validation set. The training time is similar to our proposed method. The total training time is between 14 and 50 hours on a single GTX 1080 Ti and 10 CPU cores.

We set the maximum number of edges $m$ equal to the maximum number of triangles of any example in the training data. For the spherically distributed point sets, $m$ is a constant that is $m = (n-4)2+4$ for $n \geq 4$. This can be seen from the fact that all points lie on the convex hull in this case. Note that the challenge lies not with finding which points lie on the convex hull, but in finding all the facets that constitute the convex hull. For the Gaussian distributed point sets, $m$ varies between different samples. For $n = 30$ most examples have $< 40$ edges, for $n = 50$ most examples have $< 50$ edges, and for $n = 100$ most examples have $< 60$ edges.

## D.3   Delaunay triangulation

The problem of Delaunay triangulation is, similar to convex hull finding a well-studied problem in computational geometry and has exact solutions in $\mathcal{O}(n \log(n))$ [40]. We consider the same learning task as Serviansky et al. [2], who frame Delaunay triangulation as mapping from a set of 2D points to the set of Delaunay edges, represented by the adjacency matrix. Note that this differs from finding the set of triangles, as an edge no longer remembers which triangles it is part of. Thus, this reduces to a set-to-graph task, instead of a set-to-hypergraph task.

**Model adaptation.**   The goal in this task is to predict the adjacency matrix of an ordinary graph – a graph consisting of edges that connect two nodes – where the number of edges are greater than the number of nodes. One could recover the adjacency matrix based on the matrix product of $\boldsymbol{I}^T \boldsymbol{I}$, by clipping all values above 1 back to 1 and setting the diagonal to 0. This approach is inefficient, since in this case the incidence matrix is actually larger than the adjacency matrix. Instead of applying our method directly, we consider a simple adaptation of our approach to the graph setting. We replace the initial set of edges with the (smaller) set of nodes and apply the same node refinements on both sets. This change results in $\boldsymbol{E}{=}\boldsymbol{V}$ for the prediction and effectively reduces the incidence matrix to an adjacency matrix, since it is computed based on all pairwise combinations of $\boldsymbol{E}$ and $\boldsymbol{V}$. We further replace the concatenation for the MLP modelling the incidence probability with a sum, to ensure that the predicted adjacency matrix is symmetric and represents an undirected graph. Two of the main design choices of our approach remain in this adaptation: Iterative refining of the complete graph with a recurrent neural network and BPTT with gradient skips. We train our model with $T{=}32$, $B{=}4$ and $N{=}4$. At test-time, an edge between two nodes exists if the adjacency value is greater than 0.5.

**Setup.**   We increase the latent dimensions to $d{=}256$, resulting in 595201 trainable parameters. This notably differs to Set2Graph, which increases the parameter count to 5918742 [2], an order of magnitude larger. The total training time is less than 9 hours on a single GTX 1080 Ti and 10 CPU cores.

### D.4 Learning higher-order edges

We demonstrated that the improved scaling behavior of our proposed method can be leveraged for tasks that are computationally out of reach for adjacency based approaches. The number of points and dimensions were chosen in conjunction, such that the corresponding adjacency tensor would require more storage than is feasible with current GPUs (available to us). For 13 points in 10 dimensions, explicitly storing the full adjacency tensor using 32-bit floating-point numbers would already require more than 500 GB. We intentionally kept the number of points and dimensions low, to highlight that the asymptotic scaling issue cannot be met by hardware improvements, since small numbers already pose a problem. Note that Set2Graph already struggles with convex hull finding in 3D, where the authors report that storing 3-rd order tensors in memory is not feasible. Instead, they consider a local version of the problem and take the $k$-Nearest-Neighbors out of the set of points that are part of the convex hull, with $k = 10$. While we limited our calculation of the storage requirement to the adjacency tensor itself, a typical implementation of a neural network also requires storing the intermediate activations, further exacerbating the problem for adjacency-based approaches.

### D.5 Backprop with skips

We compare backprop with skips to TBPTT [27] with $B{=}4$ every $4$ iteration, which is the setting that is most similar to ours with regard to training time. In general, TBPTT allows for overlaps between subsequent BPTT applications, as we illustrate in Figure 5. We constrict both TBPTT and backprop with skips to a fixed memory budget, by limiting any backward pass to the most recent $B{=}4$ iterations, for $T{\in}\{16, 32\}$. The standard backprop results serve as a reference point to answer the question: "What if we apply backprop more frequently, resulting in a better approximation to the true gradients?", without necessitating a grid search over all possible hyperparameter combinations for TBPTT. The results on standard backprop appear to indicate that performance worsens when increasing the number of iterations from $16$ to $32$. We observe that applying backprop on many iterations leads to increasing gradient norms in the course of training, complicating the training process. The memory limited versions did not exhibit a similar behavior, evident from the improved performance, when increasing the iterations from $16$ to $32$.

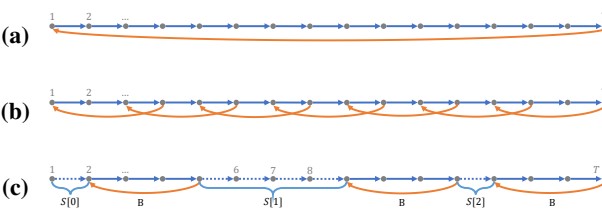

**(a)**

**(b)**

**(c)**

**Figure 5: Adding gradient skips to backprop** (a) Standard backprop (b) TBPTT, applying backprop on 4 iterations every 2nd iteration (c) Backprop with skips at iterations 1, 6, 7, 8, which effectively reduces the training time, while retaining the same number of refinement steps.

