# OpenReview forum: "Pruning Edges and Gradients to Learn Hypergraphs from Larger Sets"
_logconference.io/LOG/2022/Conference — LoG 2022 Poster_

### Official Review · Reviewer_CaxP · 2022-09-29

**Overall Score:** 5
**Confidence:** 2

**Review:**

This paper proposes a new model for the set-to-hypergraph problem, which aids in enhancing the asymptotic scaling of the problem in comparison to the current state of the art. Using the incidence matrix instead of the adjacency matrix to represent hypergraphs allows for a reduction in memory consumption. Multiple experiments were conducted to show that the method is empirically superior to the baselines.


It is difficult for me to defend the work because the paper is poorly written. Specifically, numerous terminologies lack clarity. After reading the entire paper, I'm still unsure of 1) what concrete optimization problem this work addresses, 2) what the training and testing inputs/outputs are, and 3) how function f is constructed.

Below are my detailed comments:

Intro:
line 37: “...... by pruning the non-existing edges”: The phrase "pruning the non-existing edges" is used without explanation, so readers won't understand the steps involved.

Line 38: “We prove that during training it suffices to supervise the existing edges only…”:
This sentence is unclear because line 26 states, "we start with a set of nodes without any edges." Thus, I do not comprehend the considered setting.

Line 45: “iterative refinement on a pruned hypergraph”: Again, what’s the pruned hypergraph?

Preliminary:
Line 53: “we treat the input set as the set of nodes of a hypergraph and aim to learn a function f that predicts the corresponding set of edges”: This sentence does not adequately define the problem in a formal sense. I would suggest explicitly stating the problem's inputs, outputs, and objectives to clarify matters.

Line 57: “In Set2Graph, f is split into a collection of set-to-k-edge functions Fk”: What does it mean to split into? Does this term refer to a linear combination? please provide clarification.

Section 2:
Line 69: “The goal is to learn a model f(X) = H that maps a set X of input vectors to the hypergraph H”: The term "hypergraph" requires more precise definitions. For instance, will the weights of each hyperedge be learned?

Line 70: “The choice of how to represent the hypergraph H can already drastically impact the asymptotic complexity of f, as we saw for Set2Graph: Please explain how Set2Graph represents the hypergraph, as readers are not expected to be familiar with all of its nuances.

Line 71: In what follows, we explain how we represent the nodes and edges of H to learn a pruned incidence matrix”: Again, a term without a definition. What is the meaning of "pruned incidence matrix"?

Line 76: “expect the latter to be represented as latent vectors $v \in V$ of $d_V$ dimensions”: Please define V.

Line 101: “inference time to get the discrete incidence matrix I’”: what is I’, and what’s the difference between I’ and I?

Line 105: “This highlights the importance of the latent node and edge representations, which enable us to model the dependencies in the output while still being efficient at training and inference time”: this is unclear, why does it enables efficiency?

Line 111: “When we prune all non-existing edges, learning a binary classifier in the adjacency case would no longer work due to the lack of negative examples”: Please explain negative examples, as this is mentioned multiple times but is never defined.

Line 117 and 120: “When comparing the predicted incidence matrix with a ground-truth matrix”, “We achieve this by matching every row in I with a row in the pruned ground-truth incidence matrix”: where does the ground truth matrix come from?

Importantly, Proposition 1 provides the specifics of the issue considered in this work. The authors should consider including formal definitions in the introduction to aid readers in grasping the big picture and preventing confusion.

section 3:
Line 148: “either by increasing the number of hidden dimensions or the depth of the neural network, which is clearly not scalable.”: Neural networks are introduced without any prior mention, so readers will be unaware of the context.

Line 153: “the input varies for each time step t [12]. In our case, we use the same input X at every step. “: step is not defined in the previous context.

Algorithm 1: what is S and N? Actually N is not used in the algorithm. What does L(H,I*) mean? in the previous contexts it’s L(I,I*).

Scaling by eliminating non-essential gradients: According to my understanding, this section focuses primarily on techniques to skip gradients, and the term "pruning" is not used. It would be more appropriate to name this section "skip gradients" rather than "prune gradients."

---

### Official Review · Reviewer_4tFU · 2022-10-22

**Overall Score:** 6
**Confidence:** 3

**Review:**

Summary

This paper studies set-to-hypergraph prediction, whose goal is to infer the set of relations for a given set of entities. The authors propose a model with two ideas. The first idea is to predict and supervise the positive edges only. The second idea is a training method that encourages iterative refinement of the predicted hypergraph, which allows one to skip iterations in the backward pass for improved efficiency and memory usage.

Strengths

1. The proposed ideas are clearly presented with motivations and expected advantages over existing approaches.
2. Empirical evaluation is strong. The proposed approach is evaluated on three different tasks, which are all represented as set-to-hypergraph prediction, and outperforms existing approaches consistently.
3. Ablation studies are performed extensively to demonstrate the effectiveness of the proposed ideas.

Weaknesses

1. The organization of the paper makes it hard to understand. The authors present two specific ideas (in Sec. 2 & 3) before the overall framework (in Sec. 4), making forward pointers like “The importance of this becomes clear in the discussion on the training objective later on” in line 76. This makes it hard to draw a clear picture while reading Sec. 2 and 3.
2. Comparison of running time between the proposed method and competitors is not presented, although improved efficiency is one of the main contributions of the paper.
3. The asymptotic complexity of the training and inference of the proposed approach is not presented. It would be also nice to compare the complexity with those of competitors.

Recommendation

I think this is a good paper with clear motivations, contributions, and strong empirical evaluation. I hope the authors can address the concerns for the comparison on running time or the analysis on asymptotic complexity.

Questions

1. Why are the baselines other than Set2Graph not included in Table 3?
2. N is not used in Algorithm 1
3. Can you give complexity analysis or empirical results on the running time?

---

### Official Review · Reviewer_4PU3 · 2022-10-22

**Overall Score:** 5
**Confidence:** 4

**Review:**

##########################################################################
Summary:
The authors present the problem of  set-to-hypergraph prediction which is a generalized link-prediction problem for hypergraphs. They provide solutions for 2 common issues, exponential increase in hyperedges and run time complexity. They propose to train on positive edges only and skip some backward passes to resolve these issues. Their method beats the SOTA research by a significant margin.

##########################################################################
Reasons for score:
The approach proposed in this paper is quite innovative however lacking theoretical and empirical backing. They use incidence matrix over adjacency tensors and learn the pruned incidence matrix to deal with exponential edge-blow up. They also present an iterative refinement training algorithm that operates at practically constant memory, however, not backed well. The paper is a bit hard to follow and introduces too many things all at once
##########################################################################
Pros:
The authors present an interesting direction to prevent exponentially growing number of hyperedges by training on pruned negative edges and positive edges only. Their method is clearly understandable and has good theoretical backing. The ablation study is comprehensive

##########################################################################
Cons:
For the iterative refining method, I find the theoretical backing for backward pass skips to be a little weak. It could have been explained with a better approximation estimations.
Since the paper focuses heavily on scaling, it would be nice to see resource usage and latency improvements etc.
##########################################################################
Questions during rebuttal period:
Please address and clarify the cons above
#########################################################################

---

### Official Review · Reviewer_tABr · 2022-10-22

**Overall Score:** 6
**Confidence:** 3

**Review:**

Summary:
The paper studies the problem of inferring a set of relations for a given set of entities. This problem has applications in various fields such as biological systems.  The paper proposes a method that first predicts the positive (existing) edges and thus combats the exponential running time of the brute-force algorithm. The second phase of the method proposes a training algorithm to further refine the positive edges and the third phase combines the techniques in the first two.


Reasons for the decision:

I like the overall approach to combat efficiency. However, some parts of the explanation need improvement. I have mentioned some comments and concerns. I will be happy to increase my score if the comments are addressed adequately.

Positives:

1. The considered problems are challenging especially in terms of efficiency.
2. The experiments involve two different tasks: convex hull and Delaunay triangulation.
3. The experiments show that the proposed method produces better results (Table 2 and Table 3) in different settings against different baselines.


Comments/concerns:

C1. The proposed solution feels generic and it seems a combination of several previous methods combined together. Can this part be explained further? The design of the solution is not clear with respect to the existing literature. Overall, the contribution is not clear.

C2. In the efficiency results, the main baseline Set2Graph was not included. It will be good to have a comparison of how the existing method does in terms of efficiency.

C3. Was Set2Graph [2] used with their optimized parameter for the particular Delaunay triangulation objective? Also, what is the effect of adaptation of the actual method in this case?

C4. Is it possible to show a use case? Though the experiments involve different things, the tasks are a little abstract.

---

### Meta-Review · Area_Chair_Bimm · 2022-11-21

**Confidence:** 3
**Recommendation:** Accept

**Meta Review:**

This paper proposed a set-to-hypergraph prediction framework, where the goal is to infer the set of relations among the given entities. The main idea is to predict the positive edges and then perform iterative refinement of the predicted hypergraphs. Experiments are done on two tasks, namely the convex hull and the Delaunay triangulation.

Overall reviewers found the paper to be interesting, where the efficiency gain over the brute-force algorithm is recognized by multiple reviewers. Also the reviewers found the ablation studies to be convincing. The main concerns are around the presentation, where all the four reviewers raised concerns about the clarity, the organization of the content and the notations.

There have been effective discussions during the rebuttal phase, where the authors have provided convincing explanations to most concerns. So overall based on the well motivated paper and its empirical gain on an interesting problem, I think this paper would be a good contribution to the conference. However I highly encourage the authors to take all the feedback into account and refine the paper further, to improve its quality of presentation before publication.

---

### Decision · Program_Chairs · 2022-11-23

Accept (Poster)